



# Spatial and vertical structure of precipitating clouds and the role of background dynamics during extreme precipitation event as observed by C-band Polarimetric Doppler Weather Radar at Thumba (8.50$^0$N, 77.00$^0$E)

**Kandula V Subrahmanyam and K. Kishore Kumar**

Space Physics Laboratory, Vikram Sarabhai Space Centre, Thiruvananthapuram- 695022, India

E-mail: kvsm2k@gmail.com

**Abstract**

Extreme precipitation events have been cynosure for many meteorologists as well as for common men as it causes severe weather hazards and affects the densely populated regions, especially urban cities. It is now well known that these extreme events have been increasing over the Indian region during the past few years. It becomes very important to understand and assess these events, which is challenging in terms of limited observations. Very recently, the state of Kerala, India experienced extreme rainfall events during August 2018 and led to major flooding, which is regarded as one of the worst natural disasters experienced by Kerala in the last hundred years. This catastrophic event occurred during 12$^{th}$ to 17$^{th}$ August 2018 in which the Kerala state has received 60% more rainfall than the normal during this period. The present study focuses on investigating the spatial and vertical structure of precipitating clouds and their microphysical properties during this extreme precipitation event using C-band Polarimetric Doppler Weather Radar (DWR) observations over Thumba (8.50$^0$N, 77.00$^0$E). The DWR analyses were carried out during episodes of extreme rainfall, and the time evolution of radar reflectivity structure is examined very closely to understand the structure and dynamics of this unprecedented event. The spatial and vertical structures of precipitating clouds are strongly linked with the background dynamics. Apart from the DWR observations, prevailing dynamics such as tropical easterly jet (TEJ), low-level jet (LLJ) along with vertical velocity also investigated, which showed distant signatures lead to the extreme event. It was observed that the upper level divergence existed associated with low level convergence, which aids





to the development of convection.  The westward equatorial waves were present in the period of 7-
10 days throughout the month of August 2018. The weakening of TEJ at upper troposphere resulted
in decrease of vertical shear, which favours the vertical growth of convective clouds leading to the
extreme precipitation. The enhanced strength of LLJ is also contributed to the precipitation extreme.
Thus, the significance of the present study lies in delineating the structure and dynamics of the
extreme precipitation event using indigenously developed DWR.

**Key words**: Extreme precipitation, C-band DWR, Reflectivity, Zdr, TEJ, LLJ

**1. Introduction**

India is one of the densely populated countries in the world. Due to migrations of Inter Tropical
Convergence zone (ITCZ) over the India, it experiences a strong seasonal variation in rainfall
amounts. During summer i.e., June-September (winter i.e., December-February) months of India
experiences large monsoon rains (dry period), where the ITCZ locates over the Indian subcontinent
(Ocean) (e.g., Wang 2006; Ding 2007). The monsoon rains during the summer months provides the
necessary water to the human needs as well as for societal benefits. The summer monsoon rainfall
accounts for ~80% of annual precipitation over the Indian subcontinent which is crucial for the
socioeconomic well-being and agriculture of billions of people in India (Ding and Sikka 2006;
Alcide et al. 2019). But sometimes, it can also be associated with precipitation extremes and can
impacts on the society of changing the hydrological cycle under warming climate. Even though,
there have been may advances made in understanding the role of background dynamics, which
cause extreme precipitation events and floods, but still they remain difficult predict.

Extreme precipitation events are increased in recent years and have drawn large attention across the
globe to understand their causes (e.g., Del Genio and Kovari 2002; Fowler and Kilsby 2003;



Alexander et al. 2006; Kerr 2013; Herring et al. 2014; Westra et al. 2013; Grosiman et al. 2013;
Schumacher 2019). In the past studies debated that precipitation extremes increases with warming
climate due to the larger amount of water vapour (Trenberth 1999; Allen and Ingram 2002; Pall et
al. 2007; Muller et al. 2011). Precipitation extremes are vary under climate change and the increase
in atmospheric moisture alone would able to happen precipitation extremes heavier at rate of $7\%K^{-1}$
(Held and Soden 2006; Sherwood et al. 2010). This increase in moister lead to the increase in latent
heating which may lead to stronger vertical ascent and thus adding the additional amount of
precipitation (Nie et al. 2018). The Indian summer monsoon also plays a significant role in climate
variability and previous studies confirmed that there is an increase in the amount of water vapour
over the Indian region (e.g., Mukhopadhya et al. 2017). This increase in water vapour is attributed
to the increased occurrence of extreme rainfall events over the Indian region (e.g., Goswami et al.
2006; Soden and Held 2006; Rajeevan et al. 2008). Soden and Held (2006) attributed to the large
variability in low-level monsoon jet over the Arabian Sea in recent years, which supply surplus
moisture leading to the extreme precipitation events over the Indian subcontinent. Further, the
tendency of deep convective activity is also caused for the increased water vapour in the
atmosphere. It is found that the deep convective systems, whose life time is more than 6 hours have
contributed 50% more water vapour in the mid-troposphere compared to short lived convective
systems (Baisya  et al. 2018; Derbyshire et al. 2004; Feng et al. 2012). IPCC (2002; 2007) report
emphasized to understand the behavioural characteristics of extreme rainfall events and how
background environment dynamics influences such events.

An extreme rainfall event influences the average rainfall over the region, ecosystems, land scale
through erosion processes and leads to major floods (O'Gorman 2015). It also severely affects
riverbeds and many houses were turfed by this event. Extreme precipitation events are increasing
more rather than mean precipitation in regional as well as globe and they have significant variability
across geographic locations (Alexander et al. 2006; Westra et al. 2013). Hurricane Katrina, which



was one of the most deadly storms severely, damaged the Gulf cost of America in August 2005 and
around 1800 people died (Houze et al. 2006). In 2010, a major heavy rainfall flood occurred in
Pakistan and killed around 17000 people and millions of homes were damaged (Kirsch et al. 2012).
Another major disaster happened in Canadian history in 2013 due to extreme precipitation events,
which led to Alberta floods (Milrad et al. 2015). So there is great concern about these frequency and
intensity of extreme precipitation events (Muller et al., 2011). There have been studies on extreme
rainfall trend over India (e.g., Rajendra Kumar and Dash 2001; Goswami et al. 2006; Rajeevan et
al. 2008; Ajayamohan et al. 2010; Krishnamurthy 2011) since 1998 and few of them are mentioned.
The State of Assam in India suffered by severe flood in 1998 and it regularly grappling with them.
In July 2005, Mumbai faced the very heavy rainfall, which lead to flood and killed over 700 people
and some of the areas around the Mumbai were under water. The high intensity rainfall occurred in
October 2009 was the one of the worst flood in India, which killed around 300 people and made 5
lakhs homeless. The Leh flood in August 2010, Utharkhand flood in 2013, heavy rain fall in 2014,
Gujarat flood in 2017 and very recent Kerala flood in 2018 were severely damaged the property ad
infrastructure and also took many of human life's. The extreme precipitation event occurred in
Kerala in August 2018 was the highest rainfall in a century. This event was associated with
unusually very high amount of rainfall during ISM and about 60 % more than the rainfall in Kerala
contributed by this extreme precipitation rainfall during ISM period. The west coast of Kerala and
Western Ghats receives a heavy rainfall during Indian summer monsoon region and it contributes
significantly to the average rainfall (Rao 1976; Soma and Krishnakumar 1990; George 1956;
Srinivasan et al. 1972; Mukherjee and Kumar 1976; Mukherjee et al. 1978; Mukherjee 1980). There
have been studies on the heavy rainfall events, which the daily rainfall exceeding 15 cm over the
west coast occur on many days cause extensive damage.

There have been numerous studies funnelled on numerical simulations of extreme precipitation
events (e.g., Hennessey et al. 1997; Held and Soden 2006; Muller et al. 2007; Sherwood et al. 2010;



Muller et al. 2011; Siler and Roe 2014; O'Gorman 2015; Shi and Durran 2015; Collow et al. 2016;
Nie et al. 2018; Tandon et al. 2018). Climate model studies showed that these extreme events
heavily depend on the region, where some regions experiences severe extremes, while other
experiences decrease in such events (e.g., Tandon et al. 2018). They also said that the drivers of
precipitation extremes are remains conundrum and poorly understood. Further, studies also showed
that changes in mean temperature and vertical velocity (e.g., Muller et al. 2011) and anomalous
moisture transport and lower sea level pressure (e.g., Collow et al. 2016) play a crucial role in
strength of such precipitation extremes. Orography also play a vital role in precipitation extremes
and there have studies on the understanding the orographic response to extreme precipitation events
in climate point of view (e.g., Nie et al. 2018; Muller et al. 2007; O'Gorman 2015; Shi and Durran
2015; Siler and Roe 2014; Held and Soden 2006; Sherwood et al. 2010). From their studies it found
that the occurrence of extreme precipitation events are more over lee-ward side compared to wind-
ward climatologically. This is mainly due to the downstream precipitation transport and cause the
large condensation increases, which leads to lee-ward precipitation (Siler and Roe 2014). In
addition, the changes in vertical velocity also found to be different during extreme precipitation
events and modify background thermodynamics (Shi and Durran 2015). Duration of these extreme
events impacts the amount of precipitation and also on society (Kao and Ganguly 2011). The short
bursts of extreme precipitation events might lead to an increase in total rainfall, but they do not
significantly contribute to groundwater discharge. Synoptic meteorological conditions are also led
to the development of precipitation extremes. There have been several research studies on the role
of synoptic scale influences on precipitation extremes over the globe (e.g., Catto et al. 2002;
Takahashi 2004; Catto and Pfahul 2013; Milrad et al. 2015; Agel et al. 2018). Catto et al. (2002)
studied daily rainfall over the globe and found that approximately 30-42% is related to warm fronts,
while 18-30% is related to cold fronts. However, the precipitation extremes are more likely to be
associated with warm fronts (e.g., Catto and Pfahul 2013). They found that the around 40-50%
extreme precipitation occurs nearby locations of warm fronts in the Northeast. Takahashi (2004)



studied the precipitation extreme associated with atmospheric circulation in Peru during 2002. They
suggested that heavy rainfall days are associated with the enhanced strength of low-level westerly,
which helps the development of convection by orography lifting. Agel et al. (2018) examined the
various meteorological parameters and found that there is large difference between the days with
and without extreme events. Further, they attributed three dimensional structure of tropopause is
one of the factor relevant to precipitation in terms of circulation along with other meteorological
parameters in the north east of United States. Nonetheless, these extreme precipitation events are
difficult to manage and have dire consequences on water resource, infrastructure and agriculture.

In the present study, we used C-band polarimetric Doppler Weather Radar (DWR) observations to
characterize the spatial and vertical structure of precipitating clouds during extreme precipitation
event on 15$^{th}$ August 2018. We also investigated the dynamical meteorological parameters
responsible for the occurrence of precipitation extremes using Era-interim reanalysis data. Section 2
describes the measurements of C-band DWR and soundings at Trivandrum and Era-interim
reanalysis also discussed. The temporal evolution of spatial and vertical structure of precipitating
clouds is discussed in Section 3. Section 4 presents the changes in the background dynamics. The
summary of the present study is summarized and concluding remarks were presented in section 5.
The central objective this manuscript is to characterize the spatial and vertical distribution of
reflectivity (Z), differential reflectivity (Zdr) and the role of large-scale circulation lead to the
occurrence of extreme precipitation event.

**2. Description of C-band polarimetric DWR and base products**

C-band polarimetric DWR is installed at Thumba and is continuously operating for monitoring
weather systems. This system was developed by Bharat Electronics Limited (BEL), Bangalore with
technology provided by Indian Space Research Organisation (ISRO). This is the first polarimetric



DWR installed in India along the western coast of India. Figure 1 (a) & (b) shows geo-location of
the C-band polarimetric DWR at Thumba and its building and radome of DWR can be seen. The
volume coverage pattern of C-band DWR is shown in figure 1(c). The elevation angles are $0.5^0$, $1^0$,
$2^0$, $3^0$, $4^0$, $7^0$, $9^0$, $12^0$, $15^0$, $18^0$ and $21^0$ steps. Total 11 elevation steps are considered for full volume
scan. The major specification of C-band DWR is provided in Table 1. The central operating
frequency of DWR is 5.625 GHz and the antenna rotation is 0.5 to 6 rpm (rotation per minute). The
gain of the antenna is 45 dB. The transmitter is Klystron and the peak power is 250 kW. The pulse
widths are selectable from 0.5, 1, 2, 3 and 4 μs. The antenna sweeps 360 degrees in azimuth and 11
steps in elevation as mentioned above. For the present study, we have used C-band DWR
observations during extreme precipitation event during $12^{th}$ to $18^{th}$ August 2018. During that time,
C-band DWR is operated round the clock and provided valuable information on three-dimensional
structure of precipitating clouds in extreme precipitation. The scan strategy employed in DWR as
given follows: 1. Single PRF (Pulse Repetition Frequency) - 400 Hz with 3 elevation steps; pulse
width is 2 μs; scan RPM (rotation per minute) is 1.5; DTP (dwell time pulses) is 44 and the range
resolution is 300 meters. 2. Dual PRF - 450/600 Hz with 11 elevations; pulse width is 1 μs; scan
RPM is 1.5; DTP is 57 and the range resolution is 150 meters as mentioned in Table 1. For the
present analysis, we utilized the dual PRF observations only, where the DTP is high and provides
the full volume scan. Single PRF scan takes around ~ 6 minute to complete full volume san with
three elevations, while dual PRF scan takes around ~ 8-9 minutes. Therefore, the dual PRF volume
scans are available with ~15 minutes interval time. So, there are totally ~ 8 to 9 volume scans
available in each hour. The radar data is stored in polar co-ordinates (i.e., in terms of range, azimuth
and elevation). For the present analysis, the radar data on the polar coordinates have been converted
into Cartesian coordinates.




**Table 1:** Specifications of C-band DWR at Thumba

| Major Technical specifications | |
|---|---|
| Operating frequency | 5.6 – 5.65 GHz |
| Polarization | Single and Dual polarizations |
| Antenna rotation | 0.5 to 6 rpm |
| Antenna Gain (dB) | 45 |
| Radome survival wind speed | 200 kmhr$^{-1}$ steady |
| Transmitter | Klystron |
| Transmitter peak power | 250 kW |
| Pulse width | 0.5, 1, 2, 3, 4 μs |
| Elevation levels (degree) | 0.5, 1, 2, 3, 4, 7, 9, 12, 15, 18, 21 |
| Azimuth sweeps (degree) | 0-359 |

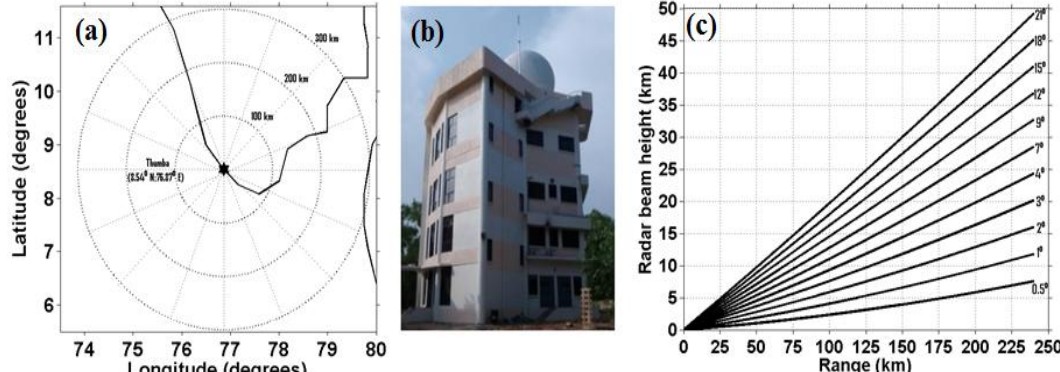


*Figure 1: Geolocation of the C-band polarimetric DWR building at Thumba and radome of DWR*
*and the volume coverage pattern*

Figure 2 (a-c) shows the base products of plan position indicator (PPI) of reflectivity (Z), Doppler
velocity (v) and spectrum width (sigma) on 15$^{th}$ August 2018 at 07:51 UT at 3 degree elevation.
Each solid black circle represents the 50 km range from the radar centre as shown in figure 2. From
this picture, it is clearly seen that the precipitating system was widely covered in the radar range and
the direction of cloud system in south-westerly as the prevailing wind is south-westerly during the
monsoon months. The maximum precipitating cloud system was concentrated in between 50 to 100
km from the radar range. The higher the reflectivity values associated with the larger rainfall
regions. The maximum reflectivity observed at this time was around 40 dBZ. The corresponding
radial velocity was shown in Figure 2(b). Radial velocity information is very useful for identifying
the change of wind with space and time. Negative (Positive) values corresponds to precipitating
cloud system is approaching towards (away) the radar. This can be clearly seen in Figure 2(b),
where blue (red) colour indicates the systems moving towards (away) the radar centre. Often the
change in wind is gradual, but it can be quite sudden in the cases of complex convective systems.
Even, a quite sudden wind change is observed during this extreme precipitation event in Figure 1(b)
indicated by red colour circle. It supposed to be negative because system is moving towards the

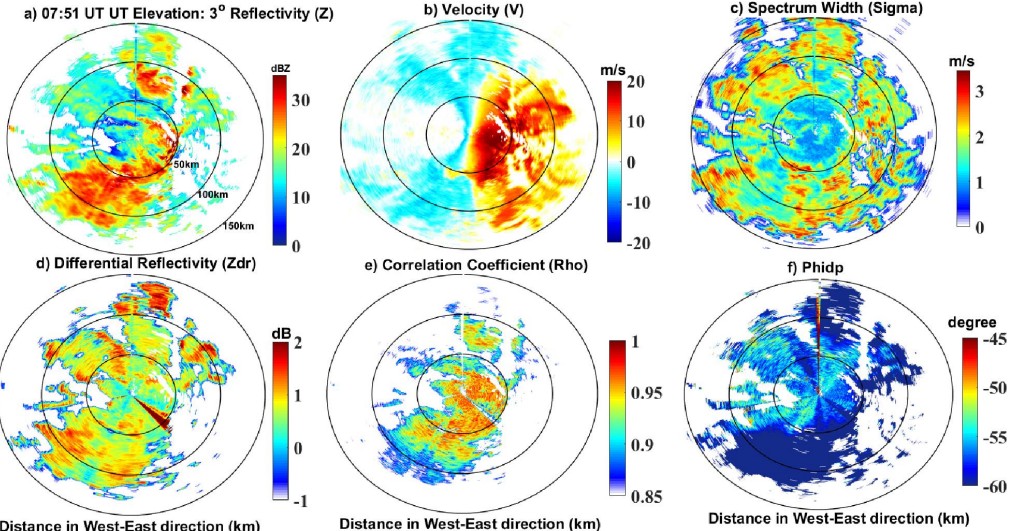


*Figure 2: C-band DWR base products (a-c): reflectivity, velocity and spectrum width; polarimetric products (e-f): differential reflectivity, correlation coefficient and differential phase shift respectively at 07:51 UT on 15$^{th}$ August 2018*

radar, but rather it is positive, which suggests that there is presence of off shore convective vortices
occurred just south of the radar location (Rao, 1976), which embedded in the extreme precipitating
system. It is also evident in Figure 2(c) of DWR spectral width where the large values of spectral



width observed. Spectral width is the measure of turbulent in precipitating systems. Figure 2(d-f)
shows the polartimetric products of differential radar reflectivity ($Z_{dr}$), cross correlation coefficient
($\rho_{hv}$) and differential phase shit ($\phi_{dp}$) respectively. The $Z_{dr}$ gives the ratio between the horizontal
reflectivity and vertical measured reflectivity. The maximum $Z_{dr}$ is found to be between 1 and 1.5
dB, which suggest that the hydrometeors are oriented in horizontal direction and oblate (Figure
2(d)). Negative $Z_{dr}$ represents the hydrometeors which are oriented vertical direction. Zero $Z_{dr}$
corresponds to the hydrometeors are circular in shape. Therefore we can identify the orientation of
the hydrometeors by inspecting distribution of $Z_{dr}$. It is evident from figure 2(d), that these echoes
mainly meteorological (Figure 2(e)) as the $\rho_{hv}$ values are 0.9 and above and are homogeneous. The
other polarimetric quantity is $\phi_{dp}$ is the difference between the phases of the copolar signal at
horizontal polarization ($\phi_{hh}$) that at the vertical polarization ($\phi_{vv}$). Around 5-10 degree phase delay
is observed during this time and this is mainly arises due to the hydrometeors in the convective
cloud systems. Since $\phi_{dp}$ is independent on radar parameters, it is very useful quantity for
calibrating the radar.

**3.      Results and Discussions**
**3.1      Temporal evolution of spatial and vertical structure of reflectivity**

Figure 3 shows the temporal evolution of spatial structure of reflectivity at 3 degree elevation from
04:30 UT to 08:00 UT with a time interval of 30 minutes on 15[th] August 2018. This picture reveals
that precipitation occurred over almost entire radar coverage. The development of precipitating
systems is clearly depicted in this figure. With time progress, enhancement of convection was
observed right from 04:30 UT to 07:51 UT in intensity as well as spatial pattern (Figure 3). The
movement of convective precipitation band is south-westerly. The maximum reflectivity is found to
be north of the radar at 04:30, 05:03 and 05:33 UT times, which is associated with deep convection,



where it was located over the Arabian Sea. It then became weak, when it moved over land at 06:04
UT and at the same times another convective system present over the ocean (south-west direction).
In addition, a distinct narrow rain band oriented along south-north direction with enhanced
reflectivity developed at 05:03 UT. Further, it widened as time progresses and intensified as seen in
Figure 3(e, f, & g). This narrow band with high reflectivity is called as monsoon warm fontal
system, which embedded in the extreme precipitating system and it indicated by black solid line in
Figure 3. The embedded monsoon frontal system enhanced with time and caused for the extreme
rainfall observed on 15th August 2018. Catto and Pfahul (2013) observed similar kind of
precipitation extreme associated with frontal systems. The precipitating system was fully covered
over the radar range by 07:51 UT. This is the time, where the most of places in and around Kerala
got heavy rainfall and it is strongly linked to the low-level westerly. During this time, the LLJ
became intensified and persisted throughout the extreme period which triggered the heavy
precipitation. Further, TEJ and vertical velocities also played vital role in enhancements of extreme
precipitation, which will be discussed in next section.

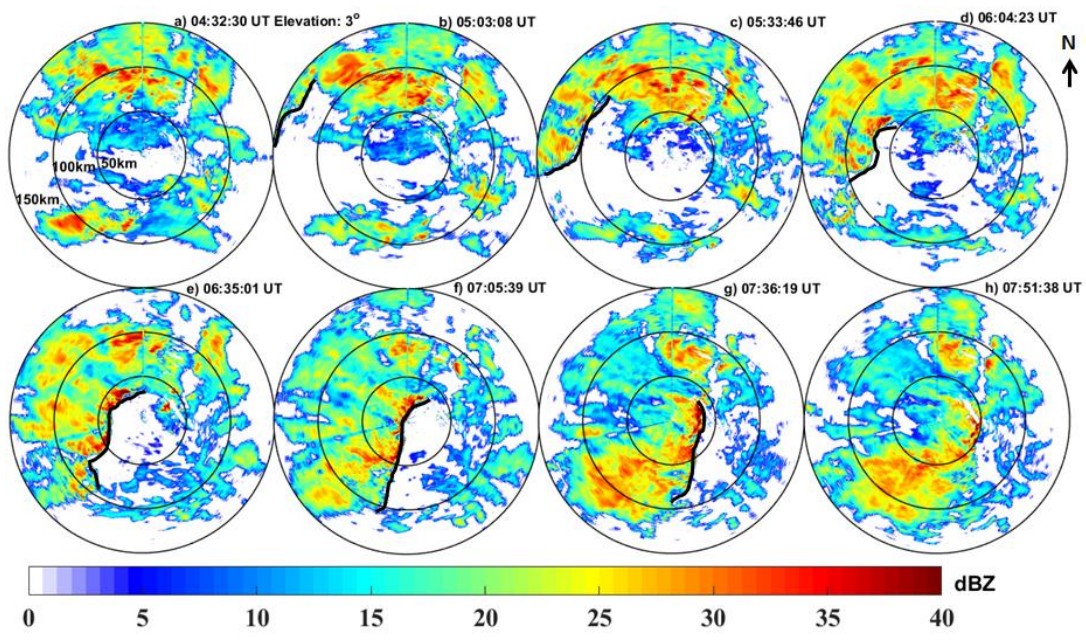




*Figure 3: (a-h): Temporal evolution of spatial structure of reflectivity at around 30 min. time*
*interval from 04:30 UT to 08:00 UT at 3 degree elevation on 15th August 2018: Black solid line*
*indicates the progresses of monsoon frontal systems.*

The precipitating cloud tops are observed to be around 10 km, which is associated with frontal
convective system as seen in Figure 4. Figure 4 shows the vertical cross sections of radar
reflectivity along frontal convective system at (a) 05:33 UT and (b) 06:35 UT. During its early
development stage, the observed precipitating cloud top is around 6 km and the convective core
width is to be around 3km (Figure 4(a)). At this time, there are multiple shallow convective cells
behind the leading convection, which are embedded in the frontal systems. Further with rapid
intensification, the precipitating cloud tops are reached beyond 10 km, where the strong updraft
prevails. And the convective core width is to be 7 km (Figure 4(b)). The maximum reflectivity is
found to be about 45 dBZ. The multiple convective cells merge and became single convective cell.
A zone of cloud free was observed immediately after the leading convective core. This is mainly
due to the descending motion associated with convective updrafts, which subsidises the cloud
development as seen Figure 4.

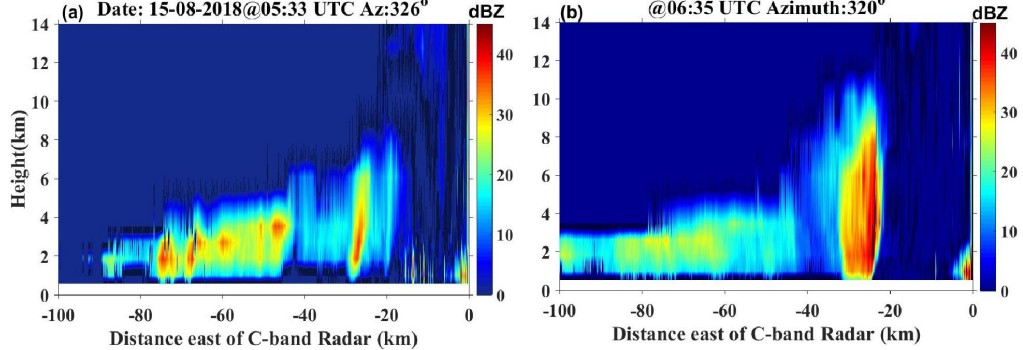


*Figure 4(a-b): Vertical cross sections of radar reflectivity along frontal convection at 05:33 and*
*06:35 UT.*

Further to study the temporal evolution of vertical profiles, time series of vertical structure of
reflectivity is plotted in Figure 5. Figure 5 shows the temporal distribution of vertical structure of
reflectivity at various development stages during extreme precipitation on 15th August 2018. At time
04:32 UT, the convective precipitating system was developed at around 20 km from the radar centre
in north-west direction as shown in Figure 5(a). With time progresses (04:32-06:05 UT), the system
was then became more vigorous with taller cumulus congests cloud with high reflectivity values
(Figure 5(b-c)).The deepest cloud observed at this time, where the higher reflectivity values are
concentrated in narrow band at 15 km west to the radar centre (Figure 5(c)). During that time
(06:05-07:05 UT), multi convective cells embedded in the extreme precipitation were observed with
broader reflectivity band as seen in Figure 5 (d-e).

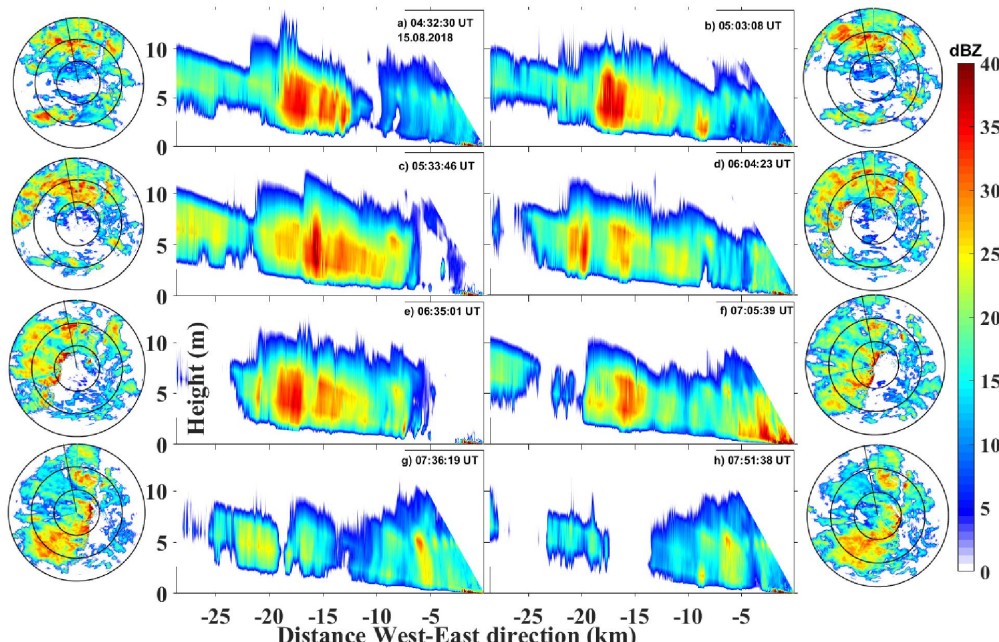


*Figure 5 (a-h): Vertical cross sections of radar reflectivity. Corresponding cross sections were also*
*shown.*

Figure 6 shows the Hovmöller diagram of (a) maximum reflectivity and (b) maximum differential
reflectivity on 15[th] August 2018 within the radar range of 80 km. Red boxes indicate the episodes of
first and second intense rainfall. From this picture, it is clearly evident that the rainfall occurred in
three spells (Figure 6(a)). The first spell was occurred during 04 UT to 08 UT, where the radar
rainfall continuously observed. And after four hours, second spell of continues rainfall occurred
from ~12 UT to 16 UT. The time duration of the third spell was less than hour. During the first spell
of extreme event, widespread rainfall with maximum Z were observed, which shows that there were
embedded convective cells with the extreme precipitation. These convective cells were eastward
propagation as clearly evident in Figure 6. Therefore, the time series of maximum reflectivity
reveals that the extreme precipitation event was not in continuous rather bursts of heavy
precipitation, which caused to receive the large amount of rainfall in a short time, resulting in a high
runoff rather than groundwater charging and subsequently led to observed flood over Kerala in
August 2018 (Mishra et al., 2018). Immediately, water levels in rivers and reservoirs have received

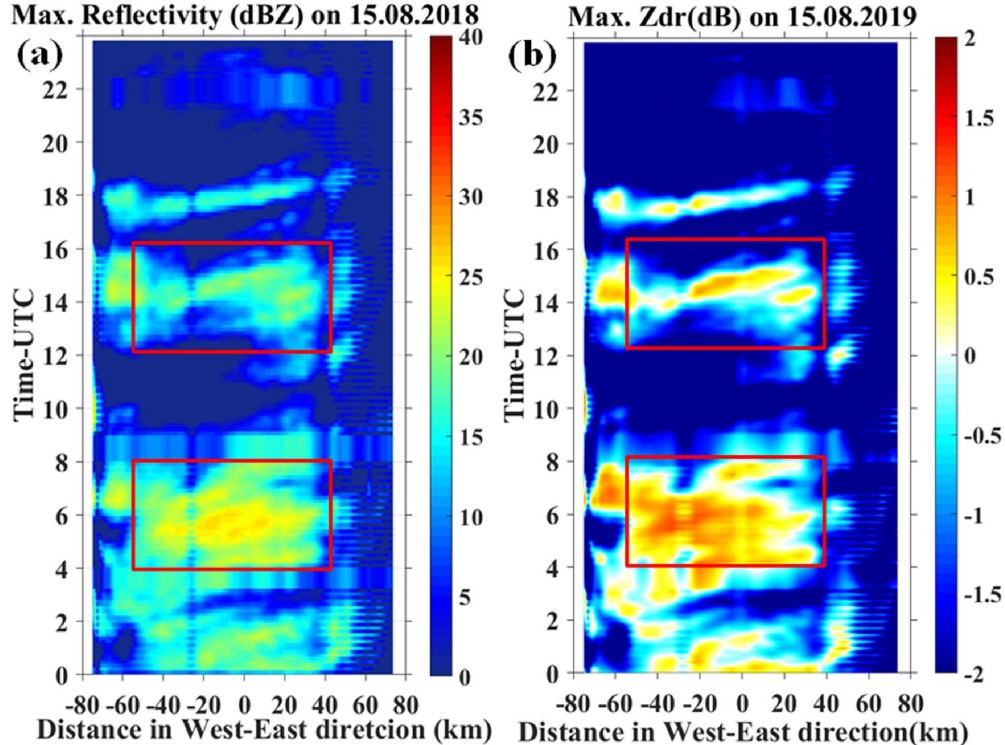


*Figure 6: Hovmöller diagram of (a) maximum reflectivity and (b) maximum differential reflectivity.*
*Red boxes indicate the episodes of intense rainfall.*

the record level of inflow. Since the DWR has the capability of polarimetry, the temporal evolution
of maximum $Z_{dr}$ shown in Figure 6(b). This provided vital information on the microphysical nature
of hydrometeor during extreme precipitation event. There were many studies using DWR



observations during extreme precipitation, but very less on polarimetric observations, especially in
India. The highest maximum $Z_{dr}$ found to be 0.8 to 1 dB, which suggests that the most of the
hydrometeor particles are oriented in horizontal with respect to radar beam and oblate in nature. The
maximum Z of ~ 18 dBZ and less value were associated with negative $Z_{dr}$, which indicates that
particle in vertically oriented (Figure 6). Figure 7 shows the DWR derived rainfall accumulation on
(a) 12$^{th}$ (b) 13$^{th}$ (c) 14$^{th}$ and (d) 15$^{th}$ August 2018. The maximum rainfall occurred during extreme
precipitation is around 300 mm and more from 12$^{th}$ to 5$^{th}$ August 2018, which led to heavy flood in
Kerala (Figure 7(d)).

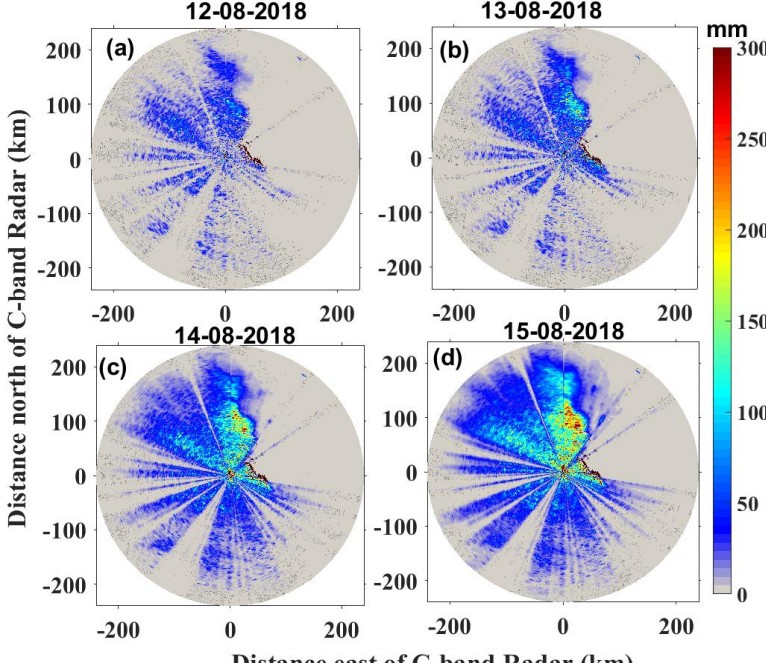


*Figure 7: Rainfall accumulation derived from C-band DWR on (a) 12$^{th}$ (b) 13$^{th}$ (c) 14$^{th}$ and (d) 15$^{th}$*
*August 2018.*

**3.2  Role of background dynamics for the observed extreme precipitation event**

In this section, we looked into the changes in the background dynamic caused for the observed
extreme precipitation in Kerala during August 2018. There should be pre-existence needed for
severe convective system to occur, which includes a high amount of convective available potential
energy (CAPE), steady supply of moisture, wind shear and low-level convergences. The Indian
monsoon is characterized by moist convective instability with relatively high amount of CAPE
(Xavier et al. 2018). Figure 8 shows that the daily variation of CAPE, which suggest that prior to
occurrence of extreme precipitation, large CAPE values observed and helps to the initiate the
vigorous convection during extreme precipitation. It is also noticed from Figure 8, that the
maximum CAPE value observed just three days priors to the day of extreme precipitation. Recently,
Xavier et al. (2018) observed the similar changes of CAPE during the Uttarakhand heavy rainfall

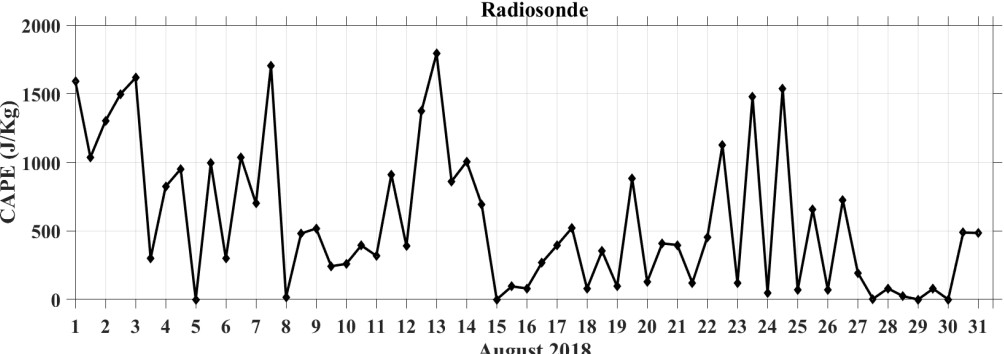


*Figure 8: Day-to-day variability of CAPE derived from radiosonde observations at Trivandrum*
*during August 2018.*

event in June 2013. We have carried analysis for three periods such as before (1st -7th August),
during (12th -18th August) and after (25th -31st August) the extreme precipitation event as shown
figures 9 & 10 for LLJ and TEJ. The Indian summer monsoon months has the strong south-westerly
wind, which carries the moisture-laden from the Arabian Sea. The supply of moisture depends on
the strength of the LLJ and orographic lifting happens when LLJ encounter Western Ghats (e.g.,
Sarker 1964). In addition, Western Ghats are also capable of producing the offshore convection and
initiate the convection due to orographic lifting (e.g., Grossman and Durran 1984; Smith 1985).
Further, to investigate the large-scale circulation, we have examined the LLJ at 850 hPa, TEJ at 150
hPa using ERA-interim reanalysis data, where the maximum zonal wind strength observed during
August 2018. In the present study, the strength of the LLJ is more during the event (Figure 9(b))
compared to before (Figure 9(a)) and after the event (Figure 9(c)). Interestingly, LLJ has single
branch during extreme event time, while it has two branches (one over the Indian continent and
other over the Srilanka) as expected in normal conditions. The enhanced moisture supply by LLJ
from the Arabian Sea into Indian subcontinent has provided the favourable conditions for
occurrence of extreme precipitation events (e.g., Priya et al. 2017). There are significant differences
in LLJ spatial structure and had a broader LLJ during extreme event compared before and after the
event (Figure 9). There have been many studies on heavy rainfall associated with increasing the
low-level winds over different parts of world (e.g., Chen and Li 1995; Takahasi 2004; Lima et al.
2009; Priya et al. 2017; Velloer et al. 2016; Xavier et al. 2018) as observed in the present case.

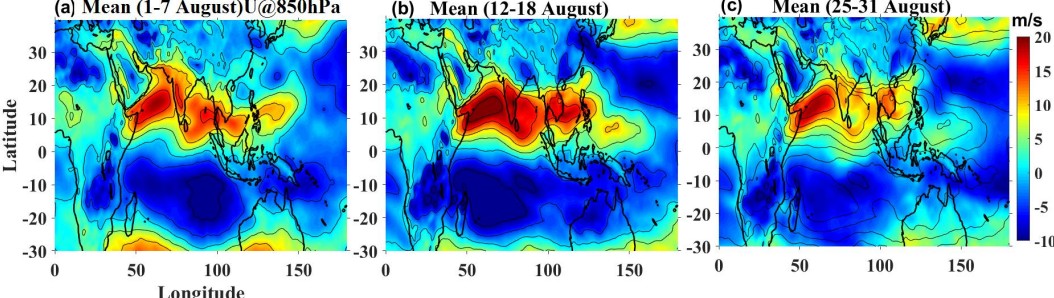


*Figure 9(a-c): The mean LLJ for three periods before (1$^{st}$ -7$^{th}$ August), during (12$^{th}$ -18$^{th}$ August)*
*and after (25$^{th}$ -31$^{st}$ August) respectively at 850 hPa.*

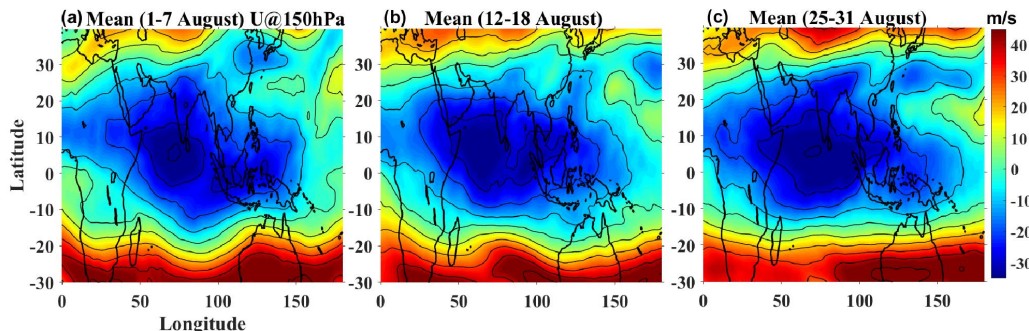


*Figure 10: Same as figure 9 but for TEJ at 150 hPa.*

We further examined the upper level changes in wind before, during and after the extreme event.
Figure 10 shows the mean upper tropospheric wind at 150 hpa (hereafter referred as TEJ) from (a)





1st -7th, (b) 12th -18th and (c) 25th -31st of August 2018 respectively over the Indian region. During
extreme precipitation period, the westerlies over the study region observed to be weakening and
further split into two as seen in Figure 10(b). Otherwise, the prevailed westerlies at 150 hPa were to
be strong as seen Figure 10 (a) and (c). The weakening of TEJ reduces the shear between the lower
troposphere and upper troposphere, which further favours the vertical development of convective
clouds as observed in spatial and vertical structure of precipitating cloud measured by C-band DWR
(Figure 4 and 5). Thus the present results indicate that the extreme precipitation event was mainly
influenced by large-scale atmospheric circulation. Chen and Li (1995) and Milrad et al. (2015) also
noted the changes in large-scale conditions were responsible for the genesis of heavy rainfall. In
modelling study by Kumar et al. (2008) also analysed the heavy rainfall event took place on 26th
July 2005 in Mumbai and found that the large-scale wind circulation motions are primary
responsible for the heavy rainfall using Weather Research Forecast model output. The changes in
meridional temperature gradient is primary responsible for the observed weakening of TEJ strength.

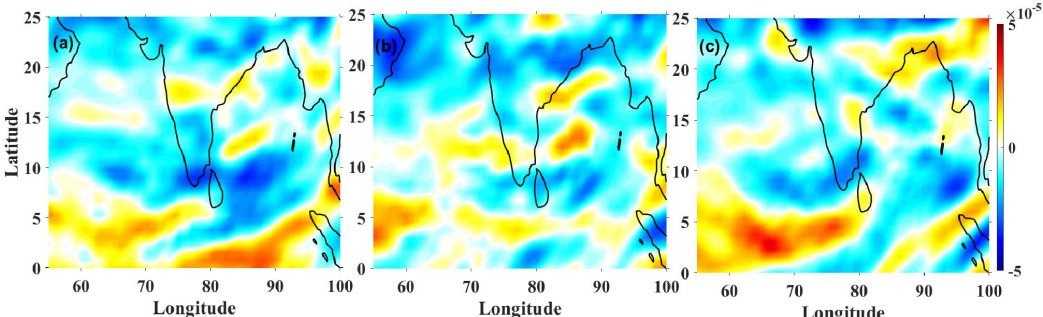


*Figure 11: Same as figure 9 but for divergence at 200 hPa.*

Figure 11 shows same as figure 9, but for divergence at 200 hPa. From this picture, it was found
that the divergence existed during the extreme precipitation period (Figure 11(b)) only, which is
associated with the low level convergence and helps to the development of convection. We also
examined the latitudinal ($5^0$-$15^0$ N) mean longitude–height structure of Omega ($\omega$) over the study
region is shown in Figure 12, which is same as Figure 9. Negative (positive) values of Omega
indicate ascending (descending) motion (Figure 12(b)) between $70^0$-$80^0$ E longitude bands. Strong





ascending was noticed from surface to 10 km height, which associated with the very intense
precipitation. Associated descending motion also enhanced during intense rainfall compared to
before and after as seen in Figure 12(a) & (c). Figure 13 shows same as figure 9, but for vertical
profile of temperature. There clear enhancement in vertical column of temperature almost
throughout troposphere during extreme event (red colour) compared to non-extreme period (black
and blue colour lines). The difference in temperature during and after was plotted in Figure 13(b). It
was found that difference in temperature around 2K from mid-troposphere (~ 4 km) to upper
troposphere (~15 km), which further fuels the development of vigorous convection as led to the
extreme precipitation. This difference mainly due to the latent heating of condensation and the
shape of temperature difference profile looks like a latent heating structure in deep convective
system with dominant stratiform region. Further, the release of latent heating during heavy
precipitation strongly influences the atmospheric circulation and feedback. The evaporative cooling
in extreme precipitation causes the negative difference in temperature at lower level (below 4km).

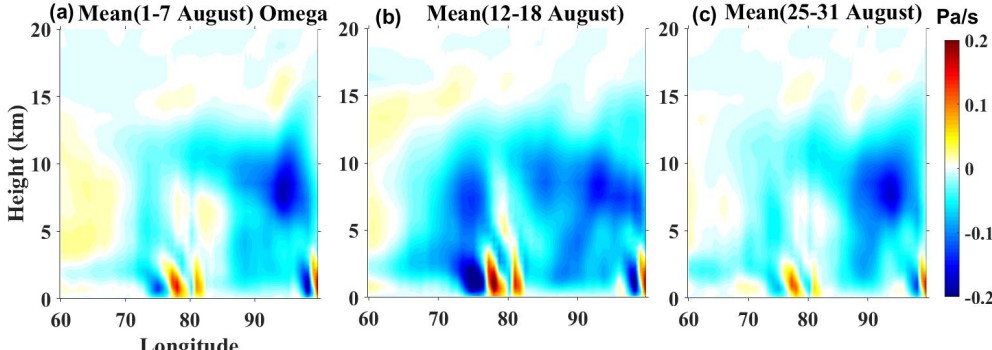


*Figure 12: same as figure 9 but for the latitudinal ($5^0$-$15^0$ N) mean longitude–height structure of*
*Omega (ω).*





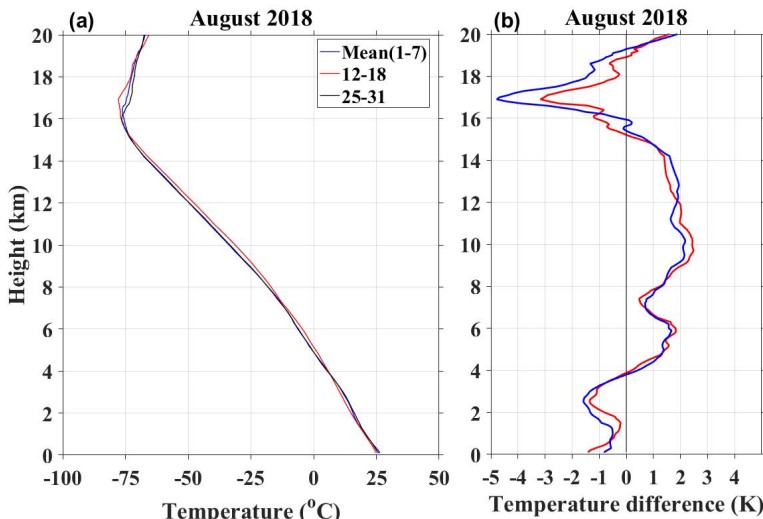


*Figure 13: (a) same as figure 9 but for Temperature profile (b) temperature difference between during and before (blue color) and during and after (red color).*

There have been few studies related extreme events to equatorially trapped waves, especially Kelvin
and Rossby waves (e.g., Wheeler et al. 2000; Takahasi 2004). We also investigated the presence of
such type of waves during extreme events. The daily wind speed anomalies over the study region at
150 hPa are subjected to wavelet analysis and extracted the period and amplitudes, which are
plotted in Figure 14. Interestingly, westward equatorial waves were present in the period of 7-10
days throughout the month of August. But their amplitude became weaken just before the extreme
event and continue to be weakened as seen in Figure 14. Wheeler et al. (2000) also found the
similar period of waves in zonal wind during heavy rainfall event and confirmed that these are
convectively coupled equatorial waves (Kelvin and Rossby) in the troposphere. Another study by
Takahasi (2004) also documented that presence of these waves. Very recently, Ferret et al. (2019)
found that the presence the probability of extreme precipitation is dependent on equatorial wave
activity and heavy precipitation can be up to three times more likely in regions of South East Asia
during the presence of equatorial waves  (Ferret et al., 2019). The weakening of equatorial waves
just before the extreme event might have caused for the observed divergence at upper level as seen
in Figure 11.Thus the present findings suggest that the combined effect of strengthened southwest



monsoon circulation, weakened westerlies at upper levels and the presence of strong updrafts were
provided the favourable conditions for the observed extreme precipitation event took place in
August 2018 over Kerala.

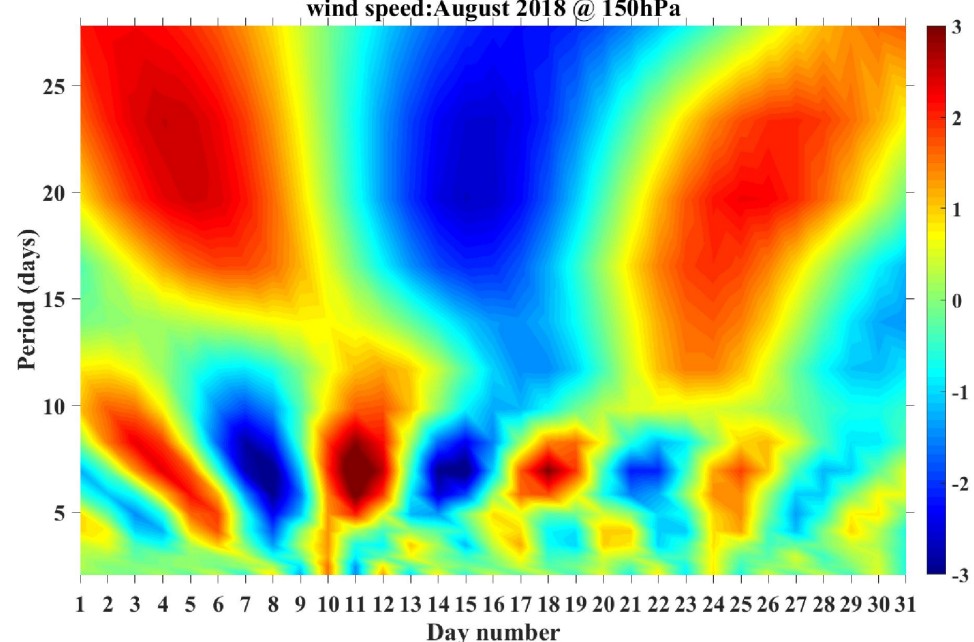


*Figure 14: Wavelet analysis for wind speed at 150 hPa on 15<sup>th</sup> August 2018.*
**4. Summary and conclusion**

The present paper investigated the spatial and vertical structure of precipitating clouds during recent
extreme precipitation event in Kerala using C-band polarimetric DWR observations at Thumba and
the role of large-scale circulations were examined. DWR observations showed that there are
multiple shallow convective cells behind the leading convective cell, which were embedded in the
frontal convective system and whose cloud tops are 10 km and above. The maximum reflectivity
and width of convective core found to be 45 dBZ and 7 km respectively. The rapid development of
precipitation took during first intense spell in which increased rain intensity as well as wide spatial
pattern was observed. The most of the hydrometeor particles are oriented in horizontal direction and



oblate in shape, where the maximum $Z_{dr}$ found to be 0.8 to 1 dB. It was found that firstly, the
atmosphere as fully saturated up to 400 hPa and has the very high CAPE which has favourable
conditions for moist convective instability to initiate occurrence of convection. It was also found
that strengthening of LLJ at lower level provided the surplus moisture and weakening of TEJ at
upper levels decrease of vertical shear, which favours the vertical development of convective
clouds. It was found that the weakening of equatorial waves (~7-10 days) during the extreme period
and might have caused the observed upper level divergence associated with low level convergence,
which helped to the development of convection. Thus, the present results suggest that the combined
effects of LLJ, TEJ and the presence of equatorial waves along with the presence of strong updrafts
were caused the occurrence of extreme precipitation. These results are summarized schematically in
Figure 15, which provided an overview of changes in background dynamics before and during the
extreme precipitation event. Hence, the significance of the present study lies in demonstrating the
role of atmospheric dynamics for the spatial and vertical structure of precipitating cloud observed
by c-band DWR during extreme precipitation event, which will be useful in understanding and
modelling the role of atmospheric circulation in extreme events.

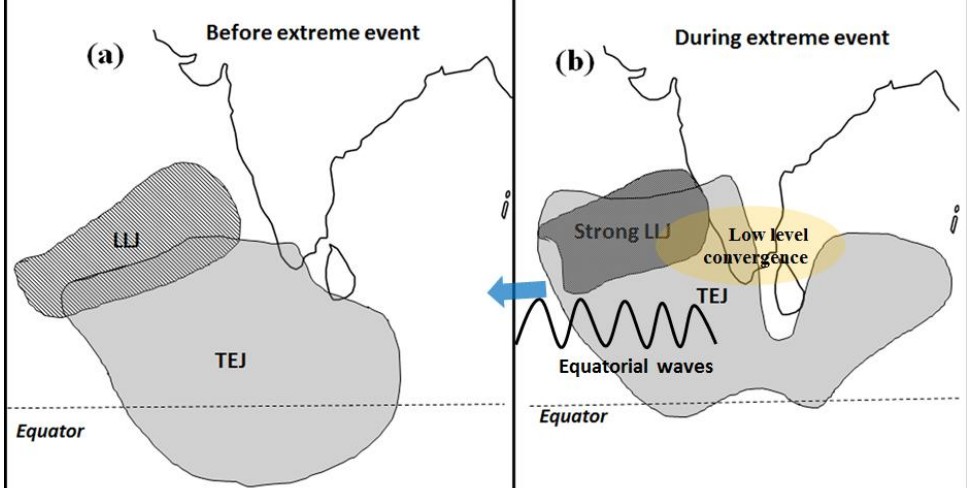


*Figure 15. Overview of changes in large-scale circulations such as spatial structure of LLJ and TEJ*
*before and during the extreme precipitation event.*



**Acknowledgements**

The soundings over Trivandrum were obtained from University of Wyoming http://weather.uwyo.edu/upperair/sounding.html. The authors thank to Era-Interim reanalysis team for providing the reanalysis winds data publically. The authors thank to DGM, TERLS and those who supported the continuous operation of DWR. We thank to Director, SPL for constant support and encouragement for carrying out this study.

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
