# Peer review of "Spatial and vertical structure of precipitating clouds and the role of background dynamics during extreme precipitation event as observed by C-band Polarimetric Doppler Weather Radar at Thumba (8.50$^0$N, 77.00$^0$E)"

_Natural Hazards and Earth System Sciences, 2020_

## Referee Comment (RC1) · Tomeu Rigo (Referee) · 6 Apr 2020

Title: **Spatial and vertical structure of precipitating clouds and the role of background dynamics during extreme precipitation event as observed by C-band Polarimetric Doppler Weather Radar at Thumba (8.500N, 77.000E)**

Author: **Kandula V Subrahmanyam**

Submitted to: **Natural Hazards and Earth System Sciences**

General comments

The author presents the capabilities of a C-band Doppler radar with dual polarimetric capabilities for analyzing the precipitating structures in a heavy rain event.

It is not necessary including the radar coordinates in the title neither in the abstract. Besides, I suggest indicating the country where the radar is place (not all the readers are familiarized with the Indian places and it allows understanding better what you will find in the manuscript)

English needs a huge reviewing. I suggest only some changes to the abstract, but there are a lot of errors in all the sections. Please, you should review the grammar.

Abstract

- "**the** cynosure" (besides, cynosure is not usual in atmospheric papers. I suggest "a focus of interest")
- Please, remove "as well as for common men"
- "This catastrophic event occurred **from** 12th to 17th August 2018 in which"
- "and the time evolution of **the** radar reflectivity structure is examined"
- "upper-level" and "lower-level" (hyphen)
- "It is well-known that these extreme events have been increasing over the Indian region during the past few years."
- I suggest rewrite as "The state of Kerala (India) experienced extreme rainfall events during August 2018. These heavy rains led to major flooding, regarded as one of the worst natural disasters experienced by the area in the last hundred years.". This is an example, but you should reduce your sentences. They are extremely long and difficult to follow.

About the structure:

- please, consider removing the lines 10-14. This is introductory and is not referring to your own work. The abstract is a trailer of your work, and you have to create interest in the readers using few sentences. All those not related to your analysis does not result interesting at this point.
- I suggest starting with the L18, and later you can present the event.
- In my opinion, what is the most interesting point of your research is that is the first time that Dual-Pol has been used in an event like this in your country. At least, I was not able of finding anything similar in the bibliography. Then, this is the key of your work and, besides the obtained results, shows that future research can improve notably the knowledge of the analyzed event type in India.
- Finally, "abbreviations should not be included without explanations" ([https://www.natural-hazards-and-earth-system-sciences.net/for_authors/manuscript_preparation.html](https://www.natural-hazards-and-earth-system-sciences.net/for_authors/manuscript_preparation.html)) and your

abstract has 345 words (an abstract of 100–200 words. https://www.natural-hazards-and-earth-system-sciences.net/about/manuscript_types.html)

Keywords

The keywords are those words that summarize your work. Do you think that the chosen ones are the correct? For instance, Monsoon or Dual-Pol provide more information.

Introduction

L69: How deep are the convective systems? Are you referring to mesoscale convective systems (MCS) or to mesoscale convective complexes (MCC) or to other type of convective mode? Can you explain this point?

L71: what IPCC means?

L96: what ISM means?

L141-144: I suggest rewriting these lines. You should introduce here the objectives (main and secondaries) of your research. However, you are explaining the analysis in general.

Description of C-band polarimetric DWR and base products

L155-156: is the radar operative for weather surveillance or is used only for research purposes?

L160-161: it looks to me that the degrees of the elevation seem "0" (zero) super index. Please, change by the correct symbol (º). Besides, if you include them in table 1, you do not need to write here.

158-180: Which is the range of the radar? Is the same for all elevations?

Table 1: the caption is not well placed

L175: san? Or scan?

Figure 1: I suggest you to remove the mid and right panels (which not provide any information), and makes the left one larger. Besides, you can improve it, including a general map of India and changing the current one by another considering the topography of the region. The new proposal would orientate the reader about the radar environment.

Figure 2: Which software have you used for displaying the radar data? Interesting to include in the text. Besides, it results interesting to explain the reason of the discontinuity in the N ray and, also, in the 120º direction (E-SE). You need to add some labels for helping the identification in the text. (L208-256)

L221: I suppose that you are referring to fig. 2

For the para that goes from L208 to 256, I suggest a re-distribution of the text. In my opinion, it is necessary that first you introduce the variable (e.g. reflectivity, radial wind, spectral width...), explaining what you analyze in the imagery, and after, a description of the image of figure 2. Besides, including the labels would help to detect the key signatures and understanding better the imagery from the point of view of the reader. In this point, you can combine with the meteorological

explanation. I suggest https://journals.ametsoc.org/doi/full/10.1175/BAMS-D-17-0317.1 for describing dual-pol variables, but there are many other in the bibliography.

Results and Discussions

 I suggest you to change the title of the section by "Analysis of the event"

Figure 3: the text about this figure must explain why you have selected the concrete period. Besides, I suggest indicating in the map what you are you referring in the text (e.g. "deep convection, where it was located over the Arabian Sea"). Why you do not indicate the line at 7.51 UT? Another thing about this time, which is important in the text: "most of places in and around Kerala", please, mark with a star or a similar symbol.

Figure 4: it would be nice to know the transect used for making the cross section (you can display it in figure 3). Besides, I think that these graphs should be accompanied of other products (radial wind or polarimetric products).

I don't understand the meaning of the figure 3 if you include after figure 4. What are you trying to explain in both figures that differentiate them? Please, you must explain clearly the intention of each figure.

Figure 5: I don't understand why are you always considering the same direction of the cross section if the system you follow is moving in time. Question about this figure and the radar functioning: did you notice about attenuation signal caused by heavy rainfall over the radome?

Figure 6: title of "y-axis" should be "direction". The description of this figure is vague and it is basic in the present manuscript. In special, the part of the polarimetry should be improved.

I do not understand the link between figure 6 and 7. In the previous cases, figures 3, 4, 5 and 6, the interaction between them was weak and need to increase some sentences explaining why you move from one subject to the other. In the case of the transition between figures 6 and 7, this transition is null. You change from polarimetric analysis of precipitation evolution to daily cumulated rainfall without explaining this move. Please, include some connectors between all the figures, being more concise in the last case.

Figure 7: the figure needs a clear improvement. There is no spatial reference (location of the area of analysis). Besides, you should explain many artifacts that appear in the imagery: effect of topography, beam blockage, propagation of the structures...

Figure 8: Please, include in some of the previous figures (preferent fig. 1) the location of the radiosonde station. This is important to know the reliability of the thermodynamic analysis on respect the area of analysis.  Besides, why do you include all the month period? Why do you not focus on the period of interest, and include the daily cumulated (or better the 12-hour cumulated) rainfall values in the area of analysis?

What is Western Ghats? For foreign people, you need to explain (and include in a map) all the geographic elements included in the study. (another example is the Arabian Sea)

Sections 3.1 and 3.2 must be better connected: you need to include some sentences explaining how meteorological aspects are related to the radar imagery.

Summary and discussions

The sentence " The maximum reflectivity 456 and width of convective core found to be 45 dBZ and 7 km respectively." makes reference to something not explained before. You cannot write here about something not shown previously.

Besides, you need to explain each conclusion in a different point.

Acknowledgements: Why are you using "the authors" if only one single person signs the manuscript?

References:

"These references have to be listed alphabetically at the end of the manuscript under the first author's name." (https://www.natural-hazards-and-earth-system-sciences.net/for_authors/manuscript_preparation.html)

I only attach some examples, but it may exist in more cases.

Allen, M. R., and Ingram, W. J.: Constraints on future changes in climate and the hydrologic cycle, Nature, 419: 224–232, 2002.

Kumar, A. D., Rotunno, R., Niyogi, D., and Mohanty, U. C.: Analysis of the 26 July 2005 heavy rain event over Mumbai, India using the Weather Research and Forecasting (WRF) model, Q. J. Royal Meteorol. Soc., 134: 1897–1910, 2008

Xavier, A., Manoj, M. G., and Mohankumar, K.: On the dynamics of an extreme rainfall event in northern India in 2013, J. Earth Syst. Sci., 127:30, https://doi.org/10.1007/s12040-018-0931-6, 2018.

Catto, J. L., Jakob, C., Berry, G., and Nicholls, N.: Relating global precipitation to atmospheric fronts, Geophys. Res. Lett., 39, L10805. https ://doi.org/10.1029/2012GL051736, 2012.

[...]

Herring, S. C., Hoerling, M. P., Peterson, T. C., and Stott, P. A. Eds.: Explaining Extreme Events of 2013 from a Climate Perspective, Bull. Amer. Meteor. Soc., 95 (9), S1–S96, 2014.

Baisya, H., Pattnaik, S., Hazra, V., Sisodiya, A., and Rai, D.: Ramifications of Atmospheric Humidity on Monsoon Depressions over the Indian Subcontinent, Scientific report, DOI:10.1038/s41598-018-28365-2, 2018.

Houze, R. A., Rasmussen, K. L., Medina, S., Brodzik, S. R., and Romatschke, U.: Anomalous Atmospheric events leading to the summer 2010 floods in Pakistan, Bull. Amer. Meteor. Soc., DOI: 10.1175/2010BAMS3173.I, 2011.

Houze, R. A., Chen, S. S., Lee, W. C., Rogers, R. F., Moore, J. A., Stossmeister, G.J., Bell, M.M., Cetrone, J.L., Zhao, W., and Brodzik, S.R.: The Hurricane Rainband and Intensity Experiment: Observations and modeling of Hurricanes Katrina, Ophelia, and Rita, Bull. Amer. Meteor. Soc., 87, 1503-1521, 2006.

Nie, J., Sobel, A. H., Shaevitz, D. A., and Wang, S.: Dynamic amplification of extreme precipitation sensitivity, Proceedings of the National Academy of Sciences of the United States of America, doi/10.1073/pnas.1800357115, 2018.

Takahashi, K.: The atmospheric circulation associated with extreme rainfall events in Piura, Peru, during the 1997–1998 and 2002 El Niño events, Anna. Geophys., 22, 3917–3926, 2004.

Kao, S. C., and Ganguly, A. R.: Intensity, duration, and frequency of precipitation extremes under 21st-century warming scenarios, J. Geophys. Res., 116, D16119, 2011

Kirsch, T. D., Wadhwani, C., Sauer, L., Doocy, S., and Catlett, C.: Impact of the 2010 Pakistan floods on rural and urban populations at six months, PLOS Currents, https://doi.org/10.1371/4fdfb212d2432, 2012.

Agel, L., Barlow, M., Colby, F., Binder, H., Catto, J. L., Hoell, A., and Cohen, J.: Dynamical analysis of extreme precipitation in the US northeast based on large-scale meteorological patterns, Clim. Dyn., https://doi.org/10.1007/s00382-018-4223-2, 2018.

Lima, K. C., Satyamurty, P., and Fernandez, J. P. R.: Large scale atmospheric conditions associated with heavy rainfall episodes in Southeast Brazil, Theor. Appl. Climatol., DOI, 10.1007/s00704-009-0207-9, 2009.

---

## Referee Comment (RC2) · Anonymous Referee #2 · 26 Jun 2020

General Comments

The work presented in this paper mainly shows that lower wind caused increased rain due to convection, which is a quite typical situation. According to the title of the paper I would expect to see flow convergence (from Doppler data) due to convection and actual polarimetric signatures (i.e. indication of particle type) in convective and precipitating
clouds (i.e. their vertical structure), but this was not done. Also, peak reflectivities are too low for extreme rainfall probably due to bad calibration of the radar. Finally, the text has a lot of grammar errors. The authors should make a careful editing of the paper whenever they want to resubmit it.

Specific Comments

l. 27: Low convergence by convection leads to upper divergence and not the opposite as it looks the way that this statement is structured. It may be due to the bad grammar which changes the meaning of many sentences in the paper. See other comments below for some of the many grammar errors in the text.

Introduction: This section is too long with many details and discussion which are not really needed and it should be shortened.

l. 42: Obviously, "dry" goes to winter season but this is not clear from the structure of the statement.

l. 58: replace "happen" with "create".

l. 60: replace "vertical ascent" with "convection" and " adding the additional" with "increasing the".

l. 63: replace "attributed to" with "connected with".

l. 65: replace "attributed to" with "examined ".

l. 67-69: delete the sentence "Further, ....in the atmosphere". It just repeats the same thing mentioned many time before.

l. 69-70: replace "have contributed" with ",correspond to".

l. 93: delete "were".

l. 169-177: put specifications in the table for short and long range operation and dual/single prf instead of discussing them in the text.

[Figure]

l. 179: describe the method to convert from polar to Cartesian coordinates. This is not that simple because the radar cell is of fixed gate length and angular width, which leads to sparse data at long ranges.

l. 223: probably the authors mean Fig. 2b, but the red circle is not visible. Also, Rho is a bit low in high rain areas, where it should be steadily above 0.95. This imply some V/H channel synchronization problem (thus, Phidp is a bit noisy too).

l. 240: The authors, state that negative Zdr represent vertically oriented (prolate) particles. Can they be more specific? There are other reasons for negative Zdr measurements, like differential noise (at edges of rain cells) or differential attenuation effects.

l. 247-248: The authors mention that Phidp is very useful for calibrating the radar. Was the radar actually calibrated with such or some other method? Did they verified it against e.g. in situ rainfall data?

Fig. 5: This not a really useful figure. Figure 4 is sufficient to show the vertical extend of the storm clouds.

l. 312: The rainfall described as "intense" corresponds to low to moderate reflectivity (rainfall rate correspondence?). Thus, it is not an intense rainfall and the time duration of core events is not many hours to result to a flood because of accumulated rainfall.

Fig. 6b: change "2019" in the title with "2018".

l. 333-334: The same comment as before for negative Zdr measurements.

Fig. 7: There is a lot of blockage (missing azimuth sectors) and ground clutter (non-regular texture of estimated rainfall field), which is strange with 11 elevations in each volume to select the one with less beam blockage and ground clutter. The 300 mm accumulated rain peaks in 4 days does not look to be a too extreme event (this depends on terrain too, but no information is provided).

Fig. 10: No wind direction is shown in Fig. 10. The authors should add wind arrows or

mention which wind component they show.
* * *

---

## Author Comment (AC1) · 4 Aug 2020

We are very much thankful to the referee for reviewing our manuscript and providing valuable suggestions. We have exactly followed the referee's instructions and revised the manuscript. We are herewith providing point-by-point response to the referee's

comments. The replies are typed in 'bold' letters.

Dear author, your research represents a very significant point in the improvement of the knowledge of Indian Monsoon dynamics, from many points of view. However, your manuscript needs a lot of changes, which are strictly necessary to done before its acceptation. General comments The author presents the capabilities of a C-band Doppler radar with dual polarimetric capabilities for analyzing the precipitating structures in a heavy rain event. It is not necessary including the radar coordinates in the title neither in the abstract. Besides, I suggest indicating the country where the radar is place (not all the readers are familiarized with the Indian places and it allows understanding better what you will find in the manuscript) English needs a huge reviewing. I suggest only some changes to the abstract, but there are a lot of errors in all the sections. Please, you should review the grammar. **We are very much thankful to the referee for providing the positive comments to further improve the manuscript. We removed the radar coordinates in title and abstract in the revised manuscript. We considered the referee's valuable suggestions and utmost care is taken for correcting the English grammar in the revised manuscript.**

Abstract "the cynosure" (besides, cynosure is not usual in atmospheric papers. I suggest "a focus of interest") **Modified in the revised manuscript**

Please, remove "as well as for common men" **Removed in the revised manuscript**

"This catastrophic event occurred from 12th to 17th August 2018 in which" **Modified in the revised manuscript**

"and the time evolution of the radar reflectivity structure is examined" **Modified in the revised manuscript**

"upper-level" and "lower-level" (hyphen) **Modified in the revised manuscript**

"It is well-known that these extreme events have been increasing over the Indian region during the past few years." **Corrected in the revised manuscript**

I suggest rewrite as "The state of Kerala (India) experienced extreme rainfall events during August 2018. These heavy rains led to major flooding, regarded as one of the worst natural disasters experienced by the area in the last hundred years." This is an example, but you should reduce your sentences. They are extremely long and difficult to follow. Modified in the revised manuscript

About the structure: Please, consider removing the lines 10-14. This is introductory and is not referring to your own work. The abstract is a trailer of your work, and you have to create interest in the readers using few sentences. All those not related to your analysis does not result interesting at this point. Following referee suggestion, we have removed those lines in the revised manuscript.

I suggest starting with the L18, and later you can present the event. Modified in the revised manuscript

In my opinion, what is the most interesting point of your research is that is the first time that Dual-Pol has been used in an event like this in your country. At least, I was not able of finding anything similar in the bibliography. Then, this is the key of your work and, besides the obtained results, shows that future research can improve notably the knowledge of the analyzed event type in India. Yes, this is the first study using the Dual-Pol observations used in our country. We are thankful to the referee for appreciating the scientific results presented in manuscript. We are now highlighted this point in the revised manuscript.

Finally, "abbreviations should not be included without explanations" (https://www.natural-hazards-and-earth-system-sciences.net/for_authors/manuscript_preparation.html) and your abstract has 345 words (an abstract of 100–200 words. https://www.natural-hazards-and-earth-system-sciences.net/about/manuscript_types.html) We apologize for this and the abstract is limited to 200 words in the revised manuscript.

Keywords The keywords are those words that summarize your work. Do you think that

the chosen ones are the correct? For instance, Monsoon or Dual-Pol provide more information. We have modified keywords as "Extreme precipitation, Monsoon, C-band DWR, Reflectivity, Dual-Pol" in the revised manuscript.

Introduction L69: How deep are the convective systems? Are you referring to mesoscale convective systems (MCS) or to mesoscale convective complexes (MCC) or to other type of convective mode? Can you explain this point? Convective systems as tall as 14 km are observed during this events. But their frequency of occurrence relatively small as compared to stratiform systems. The observed deep convective systems are mainly associated with Mesoscale Convective Complexes (MCC). These systems consist of large stratiform precipitating regions as well as anvil clouds.

L71: what IPCC means? Intergovernmental Panel on Climate Change (IPCC) and this is now included in the revised manuscript.

L96: what ISM means? Indian Summer Monsoon (ISM) and this is now included in the revised manuscript.

L141-144: I suggest rewriting these lines. You should introduce here the objectives (main and secondaries) of your research. However, you are explaining the analysis in general. Description of C-band polarimetric DWR and base products. We have introduced main objective of the present study and then explained the analysis in the revised manuscript as suggested by the referee.

L155-156: is the radar operative for weather surveillance or is used only for research purposes? The radar is operative for weather surveillance. However, the radar operation is limited to day time on most of the days except during meteorological events of interest such as cyclones, monsoon onset and extreme weather events.

L160-161: it looks to me that the degrees of the elevation seem "0" (zero) super index. Please, change by the correct symbol (o). Besides, if you include them in table 1, you do not need to write here. Corrected in the revised manuscript

158-180: Which is the range of the radar? Is the same for all elevations? The Range of the radar 240 km for pulse width of 1 micro second and it is not the same for all elevations.

Table 1: the caption is not well placed Corrected in the revised manuscript

L175: san? Or scan? It is 'scan'. We apologize for the mistake and now corrected in the revised manuscript.

Figure 1: I suggest you to remove the mid and right panels (which not provide any information), and makes the left one larger. Besides, you can improve it, including a general map of India and changing the current one by another considering the topography of the region. The new proposal would orientate the reader about the radar environment. As suggested by the referee, we have removed the mid and right panel in Figure 1 and included topography of the region in the revised manuscript.

Figure 2: Which software have you used for displaying the radar data? Interesting to include in the text. Besides, it results interesting to explain the reason of the discontinuity in the N ray and, also, in the 120o direction (E-SE). You need to add some labels for helping the identification in the text. (L208-256) We have used 'MatLab' software for processing and displaying the DWR data in this paper. We also added proper labels in the figure in the revised manuscript.

L221: I suppose that you are referring to fig. 2 Yes and we apologize for the mistake.

For the para that goes from L208 to 256, I suggest a re-distribution of the text. In my opinion, it is necessary that first you introduce the variable (e.g. reflectivity, radial wind, spectral width...), explaining what you analyze in the imagery, and after, a description of the image of figure 2. Besides, including the labels would help to detect the key signatures and understanding better the imagery from the point of view of the reader. In this point, you can combine with the meteorological explanation. I suggest https://journals.ametsoc.org/doi/full/10.1175/BAMS-D-17-0317.1 for describing dual-pol variables, but there are many other in the bibliography. As suggested by referee, we have described the DWR parameters such as reflectivity, radial wind and spectral width along with polarimetric variables (Zdr, Phidp and Rho) in the revised manuscript. Further, we also included the proper labels for the better understanding of the figures in the revised manuscript as suggested by referee.

Results and Discussions I suggest you to change the title of the section by "Analysis of the event" Changed in the revised manuscript

Figure 3: the text about this figure must explain why you have selected the concrete period. Besides, I suggest indicating in the map what you are you referring in the text (e.g. "deep convection, where it was located over the Arabian Sea"). Why you do not indicate the line at 7.51 UT? Another thing about this time, which is important in the text: "most of places in and around Kerala", please, mark with a star or a similar symbol. We have now included the map in the figure with proper labels. Also mentioned the time period in the revised manuscript

Figure 4: it would be nice to know the transect used for making the cross section (you can display it in figure 3). Besides, I think that these graphs should be accompanied of other products (radial wind or polarimetric products).I don't understand the meaning of the figure 3 if you include after figure 4. What are you trying to explain in both figures that differentiate them? Please, you must explain clearly the intention of each figure. We have now added transects used for cross section. We also included cross section of radial wind along with reflectivity maps in the revised manuscript. Figure 3 represents the temporal evolution of spatial structure of precipitating clouds. It provides the information on rapid development of clouds, while Figure 4 provides range-height intensity at two fixed azimuth (320 and 326 degrees).

Figure 5: I don't understand why are you always considering the same direction of the cross section if the system you follow is moving in time. Question about this figure and the radar functioning: did you notice about attenuation signal caused by heavy rainfall

over the radome? Figure 5 is removed in the revised manuscript as suggested by other reviewer. We did not see any significant attenuation caused by heavy rainfall over the radome.

Figure 6: title of "y-axis" should be "direction". The description of this figure is vague and it is basic in the present manuscript. In special, the part of the polarimetry should be improved. We have modified figure6 in the revised manuscript. We also added description on polarimetric parameter Zdr(differential radar reflectivity).

I do not understand the link between figure 6 and 7. In the previous cases, figures 3, 4, 5 and 6, the interaction between them was weak and need to increase some sentences explaining why you move from one subject to the other. In the case of the transition between figures 6 and 7, this transition is null. You change from polarimetric analysis of precipitation evolution to daily cumulated rainfall without explaining this move. Please, include some connectors between all the figures, being more concise in the last case. We have now included the connection and modified the text in the revised manuscript as suggested by the referee.

Figure 7: the figure needs a clear improvement. There is no spatial reference (location of the area of analysis). Besides, you should explain many artifacts that appear in the imagery: effect of topography, beam blockage, propagation of the structures... We have improved the figure as well as its description as suggested by the referee.

Figure 8: Please, include in some of the previous figures (referent fig. 1) the location of the radiosonde station. This is important to know the reliability of the thermodynamic analysis on respect the area of analysis. Besides, why do you include all the month period? Why do you not focus on the period of interest, and include the daily cumulated (or better the 12-hour cumulated) rainfall values in the area of analysis? What is Western Ghats? For foreign people, you need to explain (and include in a map) all the geographic elements included in the study. (another example is the Arabian Sea) We have modified the figures with geographic map and provided the labels in the revised
manuscript. We have also included the location of the radiosonde in the figure. We have now focused on the extreme precipitation period, i.e., 12-18, August 2018 and included the daily cumulated rainfall in the present study.

Sections 3.1 and 3.2 must be better connected: you need to include some sentences explaining how meteorological aspects are related to the radar imagery. Included in the revised manuscript

Summary and discussions The sentence "The maximum reflectivity and width of convective core found to be 45 dBZ and 7km respectively." makes reference to something not explained before. You cannot write here about something not shown previously. Besides, you need to explain each conclusion in a different point. We have now discussed this results in section 3 and the conclusions are presented point-wise.

Acknowledgements: Why are you using "the authors" if only one single person signs the manuscript? I have added an author with the approval of handling editor, before the paper was in open discussion.

References: "These references have to be listed alphabetically at the end of the manuscript under the first author's name." I only attach some examples, but it may exist in more cases. We apologize for this and stand corrected in the revised manuscript.

Please also note the supplement to this comment:
https://nhess.copernicus.org/preprints/nhess-2020-2/nhess-2020-2-AC1-supplement.pdf

---

## Author Comment (AC2) · 4 Aug 2020

We are very much thankful to the referee for reviewing our manuscript and providing valuable suggestions. We have exactly followed the referee's instructions and revised the manuscript. We are herewith providing point-by-point response to the referee's

comments. The replies are typed in 'bold' letters.

General Comments The work presented in this paper mainly shows that lower wind caused increased rain due to convection, which is a quite typical situation. According to the title of the paper I would expect to see flow convergence (from Doppler data) due to convection and actual polarimetric signatures (i.e. indication of particle type) in convective and precipitating clouds (i.e. their vertical structure), but this was not done. Also, peak reflectivities are too low for extreme rainfall probably due to bad calibration of the radar. Finally, the text has a lot of grammar errors. The authors should make a careful editing of the paper whenever they want to resubmit it.

We have included the convergence flow in the revised manuscript. We considered the referee's valuable suggestions and utmost care is taken for correcting the English grammar in the revised manuscript

Specific Comments l. 27: Low convergence by convection leads to upper divergence and not the opposite as it looks the way that this statement is structured. It may be due to the bad grammar which changes the meaning of many sentences in the paper. See other comments below for some of the many grammar errors in the text.

Corrected in the revised manuscript

Introduction: This section is too long with many details and discussion which are not really needed and it should be shortened.

We have shortened this section in the revised manuscript

l. 42: Obviously, "dry" goes to winter season but this is not clear from the structure of the statement.

Rewritten this sentence in the revised manuscript

l. 58: replace "happen" with "create".

Corrected in the revised manuscript
l. 60: replace "vertical ascent" with "convection" and " adding the additional" with "increasing the".

Corrected in the revised manuscript

l. 63: replace "attributed to" with "connected with".

Corrected in the revised manuscript

l. 65: replace "attributed to" with "examined ".

Corrected in the revised manuscript

l. 67-69: delete the sentence "Further, ....in the atmosphere". It just repeats the same thing mentioned many time before.

Corrected in the revised manuscript

l. 69-70: replace "have contributed" with ",correspond to".

Corrected in the revised manuscript

l. 93: delete "were".

Deleted in the revised manuscript

l. 169-177: put specifications in the table for short and long range operation and dual/single prf instead of discussing them in the text.

We have included the scan strategy details in the table

l. 179: describe the method to convert from polar to Cartesian coordinates. This is not that simple because the radar cell is of fixed gate length and angular width, which leads to sparse data at long ranges.

We have described the method in the revised manuscript

l. 223: probably the authors mean Fig. 2b, but the red circle is not visible. Also, Rho

is a bit low in high rain areas, where it should be steadily above 0.95. This imply some V/H channel synchronization problem (thus, Phidp is a bit noisy too).

Yes, it is for Figure 2 b, we apologize for this and corrected in the revised manuscript. We have also included a red circle in Figure 2. We have re-examined the Rho values and discussed accordingly in the revised manuscript.

l. 240: The authors, state that negative Zdr represent vertically oriented (prolate) particles. Can they be more specific? There are other reasons for negative Zdr measurements, like differential noise (at edges of rain cells) or differential attenuation effects.

We have modified the text and discussed the reasons behind the negative Zdr values as suggested by the referee.

l. 247-248: The authors mention that Phidp is very useful for calibrating the radar. Was the radar actually calibrated with such or some other method? Did they verified it against e.g. in situ rainfall data?

Yes, the radar used in the present study was calibrated and validated with in situ rainfall data. The detailed calibration and validation results can be found in Mishra et al. (2020). This reference is now included in the revised manuscript.

Mishra et al. (2020), First indigenously developed polarimetric C-band Doppler weather radar in India and its first hand validation results, 825-840, Journal of Electromagnetic Waves and Applications, https://doi.org/10.1080/09205071.2020.1742798

Fig. 5: This not a really useful figure. Figure 4 is sufficient to show the vertical extend of the storm clouds.

We have removed the Figure 5, as suggested by the referee in the revised manuscript

l. 312: The rainfall described as "intense" corresponds to low to moderate reflectivity (rainfall rate correspondence?). Thus, it is not an intense rainfall and the time duration of core events is not many hours to result to a flood because of accumulated rainfall.

Figure 6 is generated by taking mean along south-north direction. Due to this, the relatively large reflectivity values are averaged out. Relatively larger reflectivity (> 40 dBZ) is observed as seen in Figure 4. We have now included this aspect in the revised manuscript.

l. 333-334: The same comment as before for negative Zdr measurements.

The explanation for negative Zdr values are now included in the revised manuscript.

Fig. 6b: change "2019" in the title with "2018".

Corrected in the revised manuscript

Fig. 7: There is a lot of blockage (missing azimuth sectors) and ground clutter (non-regular texture of estimated rainfall field), which is strange with 11 elevations in each volume to select the one with less beam blockage and ground clutter. The 300 mm ac-cumulated rain peaks in 4 days does not look to be a too extreme event (this depends on terrain too, but no information is provided).

We have used lowest elevation scan for estimating the accumulated rainfall. We have now selected the 2 degree elevation with less beam blockage and ground clutter. We also provide the terrain information in the revised manuscript.

Fig. 10: No wind direction is shown in Fig. 10. The authors should add wind arrows or mention which wind component they show.

We have now added the wind arrows in Figure 10 in the revised manuscript.

Please also note the supplement to this comment:
https://nhess.copernicus.org/preprints/nhess-2020-2/nhess-2020-2-AC2-supplement.pdf